# A stochastic explanation for observed local-to-global foraging states in *Caenorhabditis elegans*

Andrew Margolis[1,2], Andrew Gordus[1,3]*

[1]Department of Biology, Johns Hopkins University, Baltimore, United States; [2]David Geffen School of Medicine, University of California, Los Angeles, United States; [3]Solomon H. Snyder Department of Neuroscience, Johns Hopkins University, Baltimore, United States

## eLife Assessment

This **valuable** paper uses a quantitative modeling approach to explore a well-studied transition in motor behavior in the nematode *C. elegans*. The authors provide **convincing** evidence that this transition, which has been interpreted as a two-state behavior, can instead be described as a process whose parameters are smoothly modulated within a single state. This finding provides insight into the relationships between latent internal states and observable behavioral states, and suggests that relatively simple neuronal mechanisms can drive behavioral sequences that appear more complex.

*For correspondence: agordus@jhu.edu

Competing interest: The authors declare that no competing interests exist.

**Abstract** Abrupt changes in behavior can often be associated with changes in underlying behavioral states. When placed off food, the foraging behavior of *C. elegans* can be described as a change between an initial local-search behavior characterized by a high rate of reorientations, followed by a global-search behavior characterized by sparse reorientations. This is commonly observed in individual worms, but when numerous worms are characterized, only about half appear to exhibit this behavior. We propose an alternative model that predicts both abrupt and continuous changes to reorientation that do not rely on behavioral states. This model is inspired by molecular dynamics modeling that defines the foraging reorientation rate as a decaying parameter. By stochastically sampling from the time interval probability distribution defined by this rate, both abrupt and gradual changes to reorientation rates can occur, matching experimentally observed results. Crucially, this model does not depend on behavioral states or information accumulation. Even though abrupt behavioral changes do occur, they are not necessarily indicative of abrupt changes in behavioral states, especially when abrupt changes are not universally observed in the population.

## Introduction

The search for food in the absence of informative sensory cues is an essential animal behavior (***Reiss and Rankin, 2021***). A foraging strategy that is observed in nearly all animals is the Area Restricted Search (ARS) (***Dorfman et al., 2022***). In ARS, animals randomly forage for food (***Codling et al., 2008***), but appetitive sensory cues (like an encounter with food) will cause the animal to restrict their search area by reorienting more frequently, thus increasing the likelihood of food encounters. Conversely, when removed from food, animals will decrease their reorientation rate to increase dispersal (***Reiss and Rankin, 2021***; ***Klein et al., 2017***). *Caenorhabditis elegans* and *Drosophila melanogaster* larvae appear to progressively increase their diffusion constant while foraging off of food by decreasing their rates of reorientation (***Klein et al., 2017***). However, in separate studies (***López-Cruz et al., 2019***;

*Calhoun et al., 2014*), individual worms appear to make an abrupt change from a high to low rate of reorientation. This behavior has been described as a switch from a local to global search strategy that relies on evidence accumulation to trigger the behavioral switch (*Calhoun et al., 2014*).

A central challenge with defining these possible state transitions is the stochastic nature of the behavior itself. Reorientations are random and follow Poisson statistics (*Pierce-Shimomura et al., 1999*; *Zhao et al., 2003*; *Stephens et al., 2011*; *Flavell et al., 2013*; *Gordus et al., 2015*; *Roberts et al., 2016*; *Iino and Yoshida, 2009*). This is very similar to the temporal behavior of individual molecules in solution (*Singh et al., 2022*). Individual molecules defined by the same reaction kinetics can stochastically produce long or short time intervals between reaction events, not because one molecule is inherently faster than the other, but because they are stochastically sampling from the same probability distribution. The diversity of times between individual worm reorientations is very similar to this. The exponentially decaying reorientation rate emerges from the average of trajectories that do not necessarily conform to this curve individually. However, even those that produce abrupt switches in reorientation rates could still emerge from a simple exponential decay strategy. Since reorientations occur stochastically, the abrupt changes in reorientation rates could simply be the result of stochastic sampling of an underlying decay phenomenon (*Srivastava et al., 2009*). Here, we show that a simple, stochastic model that does not rely on switches in behavioral state is sufficient to reproduce the reorientation kinetics observed in experimental data.

## Results

### Local-to-global behavioral transitions are inconsistently observed

In *López-Cruz et al., 2019*, the foraging behaviors of individual worms were tracked for 45 min after being removed from food. As observed previously (*Klein et al., 2017*), the reorientation rate from this study followed an exponential decay (*Figure 1a*). It was reported that roughly half the worms appeared to make a sudden single switch from local to global search as observed in *Calhoun et al., 2014*, however, the other half appeared to produce no discernible change in search strategy or exhibited multiple switches (*Figure 1b–d*). Whether or not a worm performed a single decision was defined in *López-Cruz et al., 2019* by fitting individual reorientation data to two lines using the MATLAB function *findchangepts* (*Figure 1b*; *Killick et al., 2012*). This function divides each trace into two regions that are defined by minimizing the sum of the residual squared error of two local linear regressions. The location of the transition point is varied until the total residual error attains a minimum (*Figure 1b*). A large change in slope indicates a sudden change in reorientation rate, and the intersection between the two lines determines the decision time (*López-Cruz et al., 2019*; *Figure 1b*). A worm was identified as making a sudden local-to-global foraging switch based on visually assessing the magnitude of the slope difference, therefore, the conclusion that 50% of the worms produced a single transition is fairly subjective. To evaluate this property more objectively, we assessed the distributions of slope differences ($s_1$-$s_2$) and transition times to see whether two densities were clearly distinguishable. The resulting distributions for the experimental data were continuous, with no clear boundary between deciders and non-deciders (*Figure 1e*).

Why do some worms appear to make a decision, while others do not? In aggregate, the reorientation rate decays (*Figure 1a*). Despite the population average conforming to a gradual decay, individual trajectories produce a wide diversity of trajectories which sometimes conform to an apparent drop in reorientation rate (*Figure 1c*), while others do not (*Figure 1d*). If the worms are executing a decision, this would seem to indicate only a fraction of the worms decide to switch from local to global foraging strategies, while others use a different strategy. An alternative hypothesis is that sudden changes in reorientation are random events that do not occur due to an underlying change in strategy, but because of the inherently random nature of the behavior itself (*Pierce-Shimomura et al., 1999*; *Zhao et al., 2003*; *Stephens et al., 2011*; *Flavell et al., 2013*; *Gordus et al., 2015*; *Roberts et al., 2016*; *Iino and Yoshida, 2009*).

### A stochastic model generates abrupt and gradual changes in search strategy

We tested this hypothesis by modeling individual worms by stochastic sampling of a decaying reorientation rate with the Gillespie algorithm (*Figure 2a*), a common strategy used to model the kinetics

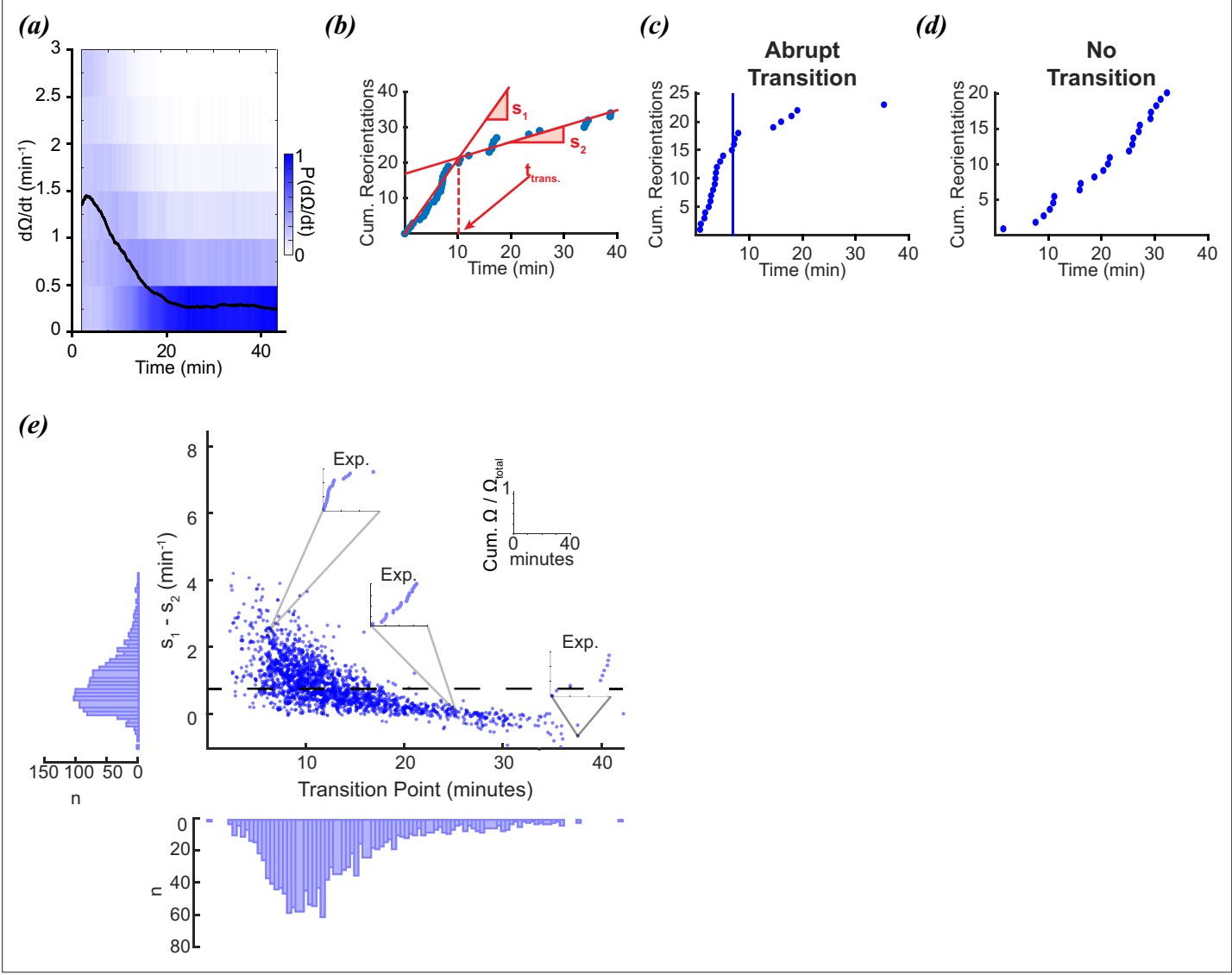

**Figure 1.** Foraging kinetics of *C.elegans*. (a) Average experimental population reorientation rate (black line) in a rolling 2 min window. Blue bins represent probability of observed reorientation rate. (b) Abrupt transitions were identified by performing two linear regressions on observed reorientation curves. Transition times ($t_{trans}$) were defined by the intersection of the regressions (dashed line). The slopes of the two regressions are $s_1$ and $s_2$. (c) An example of an experimental reorientation curve with an abrupt reorientation transition (marked by vertical line). (d) An example of an experimental reorientation curve that lacked an abrupt reorientation transition. (e) Distribution of slope differences and transition times from regressions fit to the experimental data. Individual data points are individual reorientation data for each worm. Insets are individual examples of experimental cumulative reorientation curves. Number of worms (N)=1631. Dashed line represents the median slope difference. All data curated from *López-Cruz et al., 2019*.

of individual molecules (*Gillespie, 1977*). With this strategy, the time between chemical events is modeled by randomly sampling from the time-interval distributions defined by the reaction rates. Although the algorithm was originally developed to model discrete molecular events based on known kinetic parameters, it can be used to generate time trajectories for any discrete events when the kinetics are known. A behavioral example of this is the Lotka-Volterra predator-prey competition model where predator and prey populations fluctuate out of phase due to predation. Stochastic fluctuations of predator and prey populations can be modeled using the Gillespie algorithm (*Palombi et al., 2020*; *Reichenbach et al., 2006*; *Constable and McKane, 2015*).

Experimentally, reorientation rate is measured as the number of reorientation events that occurred in an observational window. However, these are discrete stochastic events, so we can describe them

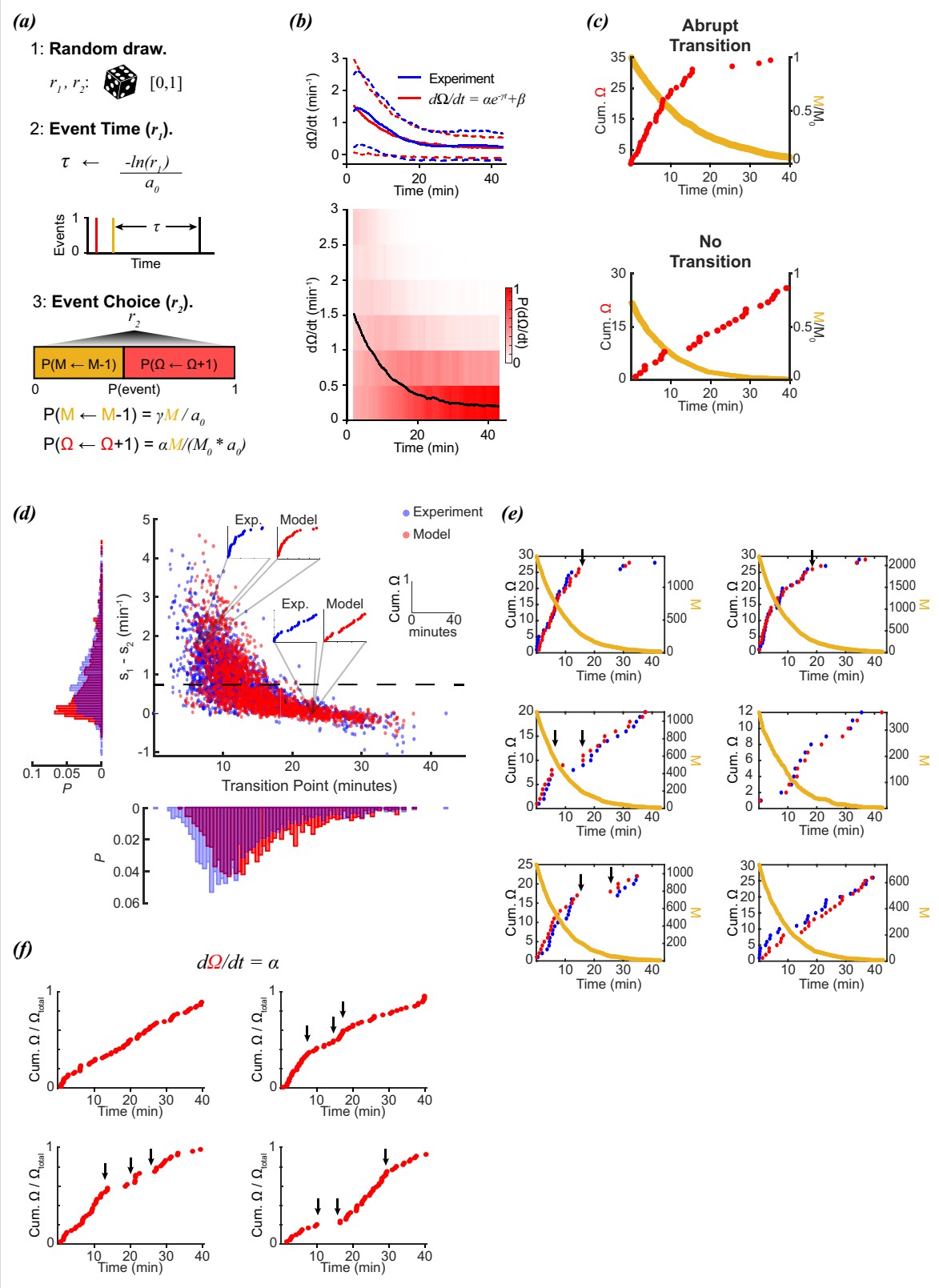

**Figure 2.** Stochastic modeling of foraging kinetics of *C. elegans*. (a) Outline of the Gillespie algorithm. Step 1: Two random numbers ($r_1$ and $r_2$) are drawn from a uniform distribution [0,1]. Step 2: The time interval for a new event ($\tau$) is randomly assigned based on the total propensities ($a_0$) and $r_1$. Step 3: The event *i* (either a decay of M (M←M-1) or a new reorientation ($\Omega$←$\Omega$+1)) at t + $\tau$ is determined by the probability (*P(i)*) of the event *i* occurring. This probability is determined by the relative propensity ($a_i/a_0$) of the event *i*. (b) (Top) The parameters $\alpha$, $\beta$, and $\gamma$ are assigned based on fitting a decay curve

*Figure 2 continued*

(red) to the observed average reorientation rate (blue). Dashed lines are + and – standard deviation. (Bottom) Average model population reorientation rate (black line) in a rolling 2 min window. Red bins represent the probability of observed reorientation rate. N=1631, $\alpha$=1.49 min$^{-1}$, $\beta$=0.1937 min$^{-1}$, $\gamma$=0.11 min$^{-1}$. (**c**) (Top) An example of a modeled abrupt reorientation transition. (Bottom) An example of a modeled reorientation curve that lacked an abrupt reorientation transition. Red data are cumulative reorientations and orange data are *M*. (**d**) Distribution of slope differences and transition times from regressions fit to the experimental (blue) and modeled (red) data. Individual data points are individual reorientation data for experimental (blue) or in silico (red) worms. Insets are individual examples of experimental and modeled cumulative reorientation curves. Dashed line represents the median experimental slope difference. (**e**) Examples of experimental (blue) and modeled (red) cumulative reorientation curves for individual worms, with similar stochastic dynamics. For each example drawn from experiments (blue), the in silico worm with the highest correlation from N=1631 iterations is shown (red). Sudden changes in rate are indicated with arrows. *M* for each example is shown in orange. (**f**) Examples of modeled data when the reorientation rate is constant ($\alpha$=1.5). Sudden changes in rate are indicated with arrows.

in terms of propensity, i.e., the probability of observing a transitional event in an infinitesimal time interval dt (in this case, a reorientation) is:

$$\frac{d\mathrm{P}\left(\Omega + 1, \mathrm{t}\right)}{dt} = a_1 P\left(\Omega, t\right) \tag{1}$$

Here, $\Omega$ is cumulative reorientation number, $P(\Omega + 1, t)$ is the probability of observing $\Omega + 1$ cumulative reorientations at time *t* (i.e. the probability of observing the number of reorientations advance from $\Omega$ to $\Omega + 1$), and $a_1$ is the propensity for this event to occur, i.e., the likelihood that a particular reaction will occur in the next infinitesimal time interval. In bulk chemical kinetics, this is analogous to the rate of the reaction. Observationally, the frequency of reorientations observed decays over time. This means that the propensity is not constant, so we can define the propensity as decaying in time:

$$a_1 = \alpha e^{-\gamma t} + \beta \tag{2}$$

Where $\alpha + \beta$ is the initial propensity at t=0, and $\beta$ is the propensity at t = $\infty$.

The propensities for the Gillespie algorithm are state-dependent; the algorithm is modeling discrete events based on the current state of the system. A time-varying propensity implies it is coupled to a state variable that is changing in time. Since the propensity $a_1$ decays exponentially, it implies it is coupled to a first-order decay process. We can model this decay as the reorientation propensity coupled to a decaying factor (M):

$$\frac{dP\left(\mathrm{M} - 1, \mathrm{t}\right)}{dt} = a_2 P\left(M, t\right) \tag{3}$$

Where the propensity of this event ($a_2$) is a first-order decay:

$$a_2 = -\gamma M \tag{4}$$

Since *M* is a first-order decay process, when integrated, the *M* observed at time *t* is:

$$M\left(t\right) = M\left(0\right) e^{-\gamma t} \tag{5}$$

We can couple the probability of observing a reorientation to this decay by redefining $a_1$ as:

$$a_1 = \frac{\alpha}{M_0} M + \beta \tag{6}$$

In this way, the propensity $a_1$ is explicitly tied to the state M, which is decaying, so that now:

$$\frac{d\mathrm{P}\left(\Omega + 1, \mathrm{t}\right)}{dt} = \left(\alpha e^{-\gamma t} + \beta\right) P\left(\Omega, t\right) \tag{7}$$

A critical detail should be noted. While reorientations are modeled as discrete events, the amount of M at time *t*=0 is chosen to be large ($M_0 \leftarrow 1{,}000$), so that over the timescale of 40 minutes, the decay in M is practically continuous. This ensures that sudden changes in reorientation rate are not due to sudden changes in M, but due to the inherent stochasticity of reorientations.

To model both processes, we can create the master equation:

$$\frac{d\mathrm{P}\left(\Omega, \mathrm{M}, t\right)}{dt} = a_1 P\left(\Omega - 1, \mathrm{M}, t\right) - a_1 P\left(\Omega, \mathrm{M}, t\right) + a_2 P\left(\Omega, M + 1, t\right) - a_2 P\left(\Omega, M, t\right) \tag{8}$$

Since these are both Poisson processes, the probability density function for a state change $i$ ($\Omega+1$: $i=1$, $M$-1: $i=2$) occurring at time $t$ is:

$$P\left(t | a_i\right) = a_i e^{-a_i t} \tag{9}$$

The probability that a state change will *not* occur for propensity $a_i$ in time interval $\tau$ is:

$$P\left(t > \tau | a_i\right) = \int_{\tau}^{\infty} P\left(t | a_i\right) dt = e^{-a_i \tau} \tag{10}$$

The probability that no state changes will occur for ALL propensities in this time interval is:

$$P\left(t > \tau | a_1\right) P\left(t > \tau | a_2\right) = \prod_i e^{-a_i \tau} = exp\left[-\tau \sum_i a_i\right] \tag{11}$$

We can draw a random number ($r_1 \in [0,1]$) that represents the probability of no events in time interval $\tau$, so that this time interval can be assigned by rearranging *Equation 11*:

$$\tau \leftarrow \frac{-ln\left(r_1\right)}{a_0} \tag{12}$$

where:

$$a_0 = \sum_i a_i \tag{13}$$

This is the time interval for any event ($\Omega+1$ or $M$-1) happening at $t + \tau$. The probability of which event occurs is proportional to its propensity:

$$P\left(event\, i\right) = \frac{a_i}{a_0} \tag{14}$$

We can draw a second number ($r_2 \in [0,1]$) that represents this probability, so that which event occurs at time t + $\tau$ is determined by the smallest $n$ that satisfies:

$$r_2 \cdot a_0 \leq \sum_{i=1}^{n} a_i \tag{15}$$

so that:

$$event\, i \leftarrow min\left\{n\, | \, r_2 \cdot a_0 \leq \sum_{i=1}^{n} a_i\right\} \tag{16}$$

For example, if the propensity $a_1$ is 10% of $a_0$ at time $t$, then there is a 10% chance that the resulting reaction will occur at time $t$. The elegant efficiency of the Gillespie algorithm is twofold. First, it models all transitions simultaneously, not separately. Second, it provides floating-point time resolution. Rather than drawing a random number, and using a cumulative probability distribution of interval times to decide whether an event occurs at discrete steps in time, the Gillespie algorithm uses this distribution to draw the interval time itself. The time resolution of the prior approach is limited by step size, whereas the Gillespie algorithm's time resolution is limited by the floating-point precision of the random number that is drawn.

In this way, the Gillespie algorithm enables the simultaneous modeling of numerous reactions in parallel with machine-precision time resolution. To ensure the reorientation rate itself is a smooth decay, a large value of M is used to approximate a continuous function. The underlying assumptions are:

1. Reorientations are stochastic (*Pierce-Shimomura et al., 1999*; *Zhao et al., 2003*; *Stephens et al., 2011*; *Flavell et al., 2013*; *Gordus et al., 2015*; *Roberts et al., 2016*; *Iino and Yoshida, 2009*).
2. All the worms experience a common decay of a signaling factor (*M*) (*Equation 5*) that influences the reorientation rate (*Equation 7*).

In our approach, we fit the exponential curve in *Equation 2* to the average reorientation rate from the experimental data from López-Cruz (*López-Cruz et al., 2019*; *Figure 2b*), and then used these parameters ($\alpha$, $\beta$, $\gamma$) to model 1631 individual worms (the same number of animals in the experimental data). Even though the initial average rates of reorientation are ~1.5 events/minute, there is considerable variance in observed initial rates (*Figure 1a*). To address this variance, starting values of *M* were assigned based on randomly drawing a rate from the observed experimental rates at $t=0$, and adjusting *M* so that the starting rate for that in silico worm *j* matched the initial rate of the randomly drawn experimental worm:

$$M_j \leftarrow \frac{M_0 \left(r - \beta\right)}{\alpha} \tag{17}$$

where *r* is an initial rate randomly drawn from the experimental data.

In silico reorientation curves generated produced single-transition and multi/absent transition curves, as observed experimentally (*Figure 2c*). When plotted along with the experimental data, the in silico data produced a distribution of linear regression parameters comparable to the experimental worms (*Figure 2d*). The simulation was able to produce individual trajectories that demonstrated switching behavior, despite the lack of a switching mechanism in the model (*Equations 2 and 3*). Furthermore, our model demonstrated a continuum of switching to non-switching behavior that was observed in experimental results (*Figure 2d*).

The experimental transition distribution was shifted slightly earlier, however, it is important to note that the experimental data were drawn from experiments where 10–15 animals were tracked together. Collisions occurred, but were excluded from analysis, and so would not contribute to this observation. However, in addition to collisions, animals also reorient in response to pheromones when they cross the tracks left by other animals, which is more likely early in the recording when the animals are in close proximity (*Hong et al., 2017*; *Dal Bello et al., 2021*). This, in turn, may contribute to a higher initial rate early in the experimental observations, which would delay the observed decay in reorientations. Since this dataset does not have information about when and where worms crossed pheromone tracks from other animals, this effect cannot be excluded.

This exception aside, modeling worm foraging behavior with a simple exponential decay of reorientations was sufficient to capture most experimentally observed dynamics, both switch and non-switch-like. Sudden switches between fast and slow reorientation rates were observed in both the experimental and modeled data (*Figure 2e*), despite the model not relying on a switch strategy, and the decay in M being continuous. Sometimes the experimental data produced switch-like behavior (*Figure 2e*, upper left panel), while other times the reorientation rate appeared to be constant (*Figure 2e*, lower right panel). The stochastic model produced reorientation data with similar dynamics, despite not relying on a decision paradigm.

It is important to emphasize that the model was not explicitly designed to match the sudden changes in reorientation rates observed in the experimental data. Kinetic parameters were simply chosen to match the *average* population behavior. Sudden changes in reorientation rate were not due to sudden changes in the underlying model; stochastic behavior naturally produces sudden bunching of random events. Even if the reorientation rate is set to a constant value, stochastic sampling will still produce sudden changes in rate, even though the rate has no time-dependence (*Figure 2f*).

## Discrete changes in M are inconsistent with experimental data

A large initial value for $M_0$ ($M_0=1000$) was initially chosen to ensure a smooth change in reorientation rate, so that sudden changes in reorientation number were not necessarily due to sudden changes in *M*. To test how sensitive the model was to stochastic changes in *M*, the initial value of *M* was tested at $M_0=1000$, 100, 10, or 1. On average, all values of $M_0$ were sufficient to reproduce the average reorientation decay observed in experimental data (*Figure 3a*). The distributions of transition times

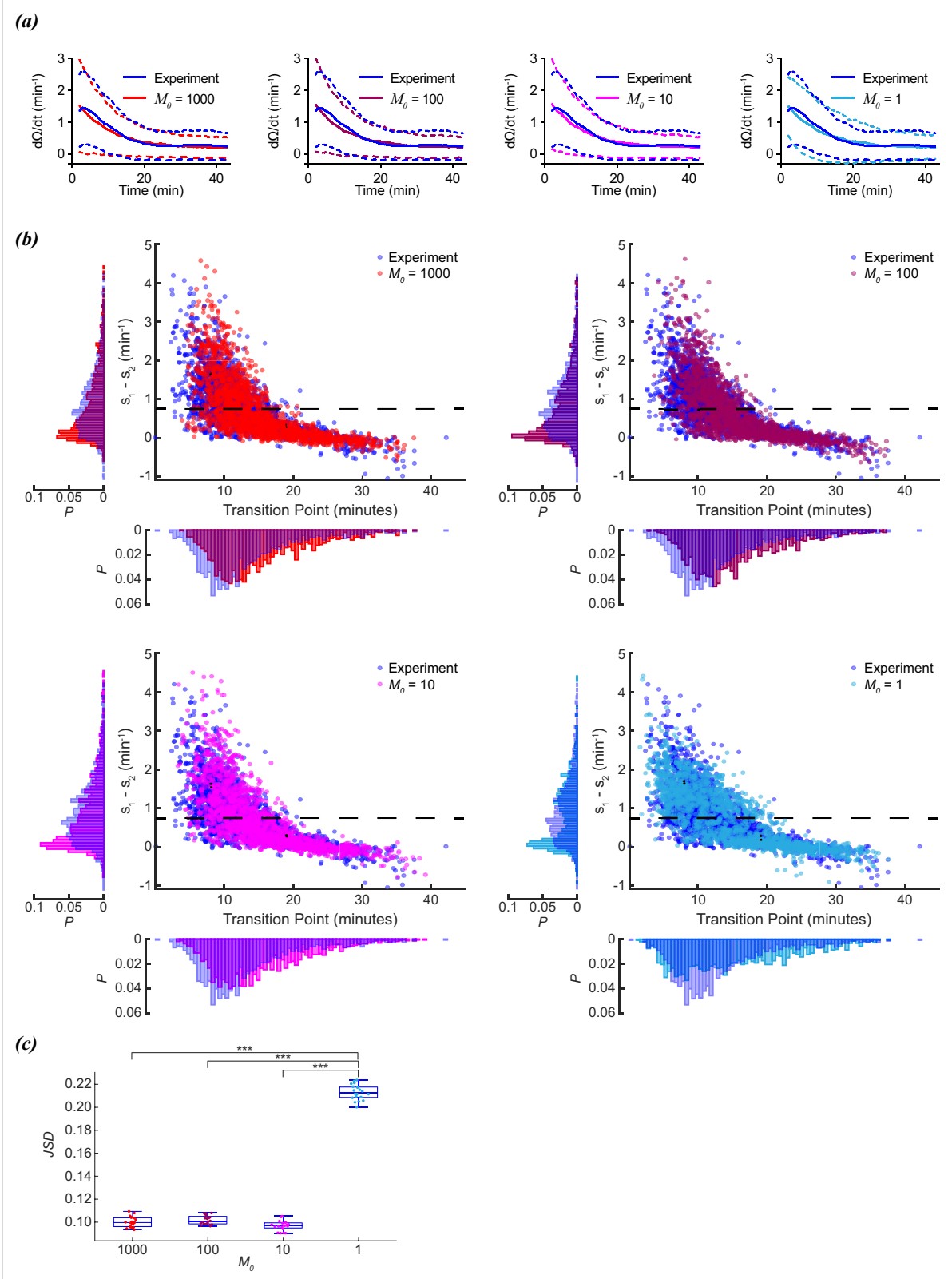

**Figure 3.** Scaling of foraging kinetics of *C. elegans*. (**a**) Average reorientation rates for experiments (blue), and models where $M_0$=1000 (red), $M_0$=100 (plum), $M_0$=10 (magenta), or $M_0$=1 (cyan). Dashed lines indicate standard deviation. (**b**) Distribution of slope differences and transition times from regressions fit to the experimental data (blue), and models where $M_0$=1000 (red), $M_0$=100 (plum), $M_0$=10 (magenta), or $M_0$=1 (cyan). Number of worms for both experiment and models (N)=1631. Dashed line represents the median slope difference for the experimental data. (**c**) The Jensen-Shannon

*Figure 3 continued on next page*

*Figure 3 continued*

divergence between the experimental distribution for slope difference and transition time in (**b**), and the distributions generated from models where $M_0$=1,000 (red), $M_0$=100 (plum), $M_0$=10 (magenta), or $M_0$=1 (cyan). Twenty distributions were generated for each M0 model. Quartiles represented by box and whiskers. P-values calculated using Tukey's range test: <0.001: ***.

and slope differences were also comparable for $M_0$=1000, 100, or 10, indicating that the model is not particularly sensitive to initial values of $M_0$.

However, despite producing a similar average decay in reorientation rates, the distribution of slope differences was distinctly bimodal for $M_0$=1 (*Figure 3b*). This was not surprising, since when $M_0$=1, the model is by definition a two-state system. Either the worm has a reorientation rate of $\alpha+\beta$ (*M*=1), or a reorientation rate of $\beta$ (*M*=0). The probability of switching from *M*=1 to *M*=0 increases with a times-cale of $\gamma$ ($\gamma$=0.11 min$^{-1}$). The bimodal slope and transition time distributions for $M_0$=1 were distinctly different from those observed for the experimental or $M_0$=1000, 100, 10 models.

Since the slope and transition time distributions were generated from a recursive algorithm rather than a purely analytical model, the model distribution for these metrics was compared to the experimental distribution using the Jensen-Shannon divergence (JSD):

$$JSD\left(P\|Q\right) = \frac{1}{2}D\left(P\|M\right) + \frac{1}{2}D\left(Q\|M\right) \tag{18}$$

where *D* is the Kullback-Leibler divergence:

$$D(P\|Q) = \sum_{x} P(x)\ln\left(\frac{P(x)}{Q(x)}\right) \tag{19}$$

and *P(x)* and *Q(x)* are the two probability distributions being compared. *M(x)* is a mixture distribution of *P* and *Q*. This metric provides a useful, parameter-free measure for comparing distributions using mutual information. A short distance between two distributions indicates they are more similar than two distributions separated by a longer distance.

To compare how similar the different models ($M_0$=1000, 100, 10, 1) compared to the experimental distribution of transition times and slope differences, we ran the model twenty times for each $M_0$, and calculated the JSD between the model and experimental distributions (*Figure 3c*). The non-binary models ($M_0\neq$ 1) all produced comparable distances, but the binary models ($M_0$=1) were significantly more distant than the rest, indicating this model was more dissimilar than the others when compared to the experimental data. A continuous decline in reorientation rate is more consistent with experimental observations than a discrete change in reorientation rate.

## Discussion

The lack of a decision simplifies the dispersal strategy for *C. elegans*. Rather than relying on the accumulation of evidence to make a discrete decision, the worm relies on a decaying signal in the absence of food that drives the reorientation rate. This strategy increases the diffusion constant of the worm, and ensures a more efficient search strategy to find food (*Klein et al., 2017*). The stochastic nature of this search is consistent with prior characterizations of worm behavior and neuronal dynamics (*Pierce-Shimomura et al., 1999*; *Zhao et al., 2003*; *Stephens et al., 2011*; *Flavell et al., 2013*; *Gordus et al., 2015*; *Roberts et al., 2016*; *Iino and Yoshida, 2009*), and implies that any individual worm would exhibit all the variability of the population if allowed to perform this task multiple times. We consider this to be a null model: simple stochastic models should be sufficient to explain observable stochastic behaviors.

The decay in M can be considered the memory timescale of the last food encounter. With an observed γ of 0.07 min$^{-1}$, this would mean a memory $\tau_{1/2}$ of ~10 min. In principle, M should rise in the presence of food to increase the reorientation rate, on a timescale that is not addressed here. In *López-Cruz et al., 2019*, the starting reorientation rate correlated with food density prior to food removal, which would be consistent with the amplitude of M correlating with food density/quality. The reliance of the reorientation rate while foraging for food on a decaying factor (M) implies that the rate is influenced by a signaling factor which decays in time. It is well established that the reorientation

rate is influenced by numerous signaling factors and is the basis of the biased random walk that drives taxis in shallow gradients (*Pierce-Shimomura et al., 1999*; *Zhao et al., 2003*; *Stephens et al., 2011*; *Flavell et al., 2013*; *Gordus et al., 2015*; *Roberts et al., 2016*; *Iino and Yoshida, 2009*).

A decay in reorientation rate, rather than a sudden change, is consistent with observations made by *López-Cruz et al., 2019*. They found that the neurons AIA and ADE redundantly suppress reorientations, and that silencing either one was sufficient to restore the large number of reorientations during early foraging. This implies that the timescale of the variable M may represent the timescales of AIA and ADE activities. AIA and ADE synapse onto several neurons that drive reorientations (*López-Cruz et al., 2019*; *Gray et al., 2005*), and their output was inhibited over long timescales (tens of minutes) by presynaptic glutamate binding to MGL-1, a slow G-protein coupled receptor expressed in AIA and ADE. Their results support a model where sensory neurons suppress the synaptic output of AIA and ADE, which in turn leads to a large number of reorientations early in foraging. As time passes, glutamatergic input from the sensory neurons decreases, which leads to disinhibition of AIA and ADE and a subsequent suppression of reorientations.

The sensory inputs into AIA and ADE are sequestered into two separate circuits, with AIA receiving chemosensory input and ADE receiving mechanosensory input. Since the suppression of either AIA or ADE is sufficient to increase reorientations, the decay in reorientations is likely due to the suppression of both of these neurons decaying in time. This correlates with an observed decrease in sensory neuron activity as well, so the timescale of reorientation decay could be tied to the timescale of sensory neuron activity, which in turn is influencing the timescale of AIA/ADE reorientation suppression. This implies that our factor 'M' is likely the sum of several different sensory inputs decaying in time.

The molecular basis of which sensory neuron signaling factors contribute to decreased AIA and ADE activity is made more complicated by the observation that the glutamatergic input provided by the sensory neurons was not essential, and that additional factors besides glutamate contribute to the signaling to AIA and ADE. In addition to this, it is not simply the sensory neuron activity that decays in time, but also the sensitivity of AIA and ADE to sensory neuron input that decays in time. Simply depolarizing sensory neurons after the animals had starved for 30 min was insufficient to rescue the reorientation rates observed earlier in the foraging assay. This observation could be due to decreased presynaptic vesicle release and/or decreased receptor localization on the postsynaptic side.

In summary, there are two neuronal properties that appear to be decaying in time. One is sensory neuron activity, and the other is decreased potentiation of presynaptic input onto AIA and ADE. Our factor 'M' is a phenomenological manifestation of these numerous decaying factors. Altered ionotropic glutamate signaling and dopamine release also influence foraging kinetics (*Hills et al., 2004*), as well as neuropeptides (*Campbell et al., 2016*). The signaling factor M could be the result of one or all of these factors decaying in time. Further work will be needed to reveal how the kinetics of reorientations emerge explicitly from these underlying signaling kinetics.

## Acknowledgements

We thank A López-Cruz and C Bargmann for sharing their experimental data, and also thank them, A Samuel and members of the Gordus lab for helpful discussions and comments on the manuscript. AG acknowledges funding from NIH (R35GM124883).

## Additional information

### Funding

| Funder | Grant reference number | Author |
| --- | --- | --- |
| National Institute of General Medical Sciences | R35GM124883 | Andrew Gordus |

The funders had no role in study design, data collection and interpretation, or the decision to submit the work for publication.

## Author contributions
Andrew Margolis, Resources, Software, Formal analysis, Validation, Investigation, Visualization, Writing – original draft, Writing – review and editing; Andrew Gordus, Conceptualization, Software, Formal analysis, Supervision, Funding acquisition, Validation, Visualization, Writing – original draft, Project administration, Writing – review and editing

## Author ORCIDs
Andrew Margolis http://orcid.org/0009-0006-9817-7147
Andrew Gordus https://orcid.org/0000-0002-5550-0286

Reviewer #1 (Public review): https://doi.org/10.7554/eLife.104972.4.sa1
Reviewer #2 (Public review): https://doi.org/10.7554/eLife.104972.4.sa2
Author response https://doi.org/10.7554/eLife.104972.4.sa3

## Data availability
Experimental data were curated from *López-Cruz et al., 2019*. Analysis was performed using custom written code in MATLAB which can be found in this repository (https://github.com/GordusLab/Margolis_foraging copy archived at *Gordus, 2025*).

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
