## [Editor Report · eLife Assessment]

This **valuable** paper uses a quantitative modeling approach to explore a well-studied transition in motor behavior in the nematode *C. elegans*. The authors provide **convincing** evidence that this transition, which has been interpreted as a two-state behavior, can instead be described as a process whose parameters are smoothly modulated within a single state. This finding provides insight into the relationships between latent internal states and observable behavioral states, and suggests that relatively simple neuronal mechanisms can drive behavioral sequences that appear more complex.

---

## [Referee Report · Reviewer #1 (Public review)]

This paper concerns mechanisms of foraging behavior in *C. elegans*. Upon removal from food, *C. elegans* first executes a stereotypical local search behavior in which it explores a small area by executing many random, undirected reversals and turns called "reorientations." If the worm fails to ﬁnd food, it transitions to a global search in which it explores larger areas by suppressing reorientations and executing long forward runs (Hills et al., 2004). At the population level, reorientation rate declines gradually. Nevertheless, about 50% of individual worms appear to exhibit an abrupt transition between local and global search, which is evident as a discrete transition from high to low reorientation rate (Lopez-Cruz et al., 2019). This observation has given rise to the hypothesis that local and global search correspond to separate internal states with the possibility of sudden transitions between them (Calhoun et al., 2014). The objective of the paper is to demonstrate that is not necessary to posit distinct internal states to account for discrete transitions from high to low reorientation rate. On the contrary, discrete transitions can occur simply because of the stochastic nature of the reorientation behavior itself.

Major strengths and weaknesses of the methods and results

The model was not explicitly designed to match the sudden, stable changes in reorientation rates observed in the experimental data from individual worms. Kinetic parameters were simply chosen to match the average population behavior. Nevertheless, many sudden stable changes in reorientation rates occurred. This is a strong argument that apparent state changes can arise as an epiphenomenon of stochastic processes.

The new stochastic model is more parsimonious than reorientation-state change model because it posits one state rather than two.

A prominent feature of the empirical data is that 50% of the worms exhibit a single (apparent) state change and the rest show either no state changes or multiple state changes. Does the model reproduce these proportions? This obvious question was not addressed.

There is no obvious candidate for the neuronal basis of the decaying factor M. The authors speculate that decreasing sensory neuron activity might be the correlate of M but then provide contradictory evidence that seems to undermine that hypothesis. The absence of a plausible neuronal correlate of M weakens the case for the model.

Appraisal of whether the authors achieved their aims, and whether the results support their conclusions

The authors have made a convincing case that is not necessary to posit distinct internal states to account for discrete transitions from high to low reorientation rate. On the contrary, discrete transitions can occur simply because of the stochastic nature of the reorientation behavior itself.

Impact of the work on the field, and the utility of the methods and data to the community

Posting hidden internal states to explain behavioral sequences is gaining acceptance in behavioral neuroscience. The likely impact of the paper is to establish a compelling example of how statistical reasoning can reduce the number of hidden states to achieve models that are more parsimonious.

---

## [Referee Report · Reviewer #2 (Public review)]

Summary:

In this study, the authors build a statistical model that stochastically samples from a time-interval distribution of reorientation rates. The form of the distribution is extracted from a large array of behavioral data, is then used to describe not only the dynamics of individual worms (including the inter-individual variability in behavior), but also the aggregate population behavior. The authors note that the model does not require any assumptions about behavioral state transitions, or evidence accumulation, as has been done previously, but rather that the stochastic nature of behavior is "simply the product of stochastic sampling from an exponential function".

Strengths:

This model provides a strong juxtaposition to other foraging models in the worm. Rather than evoking a behavioral transition function (that might arise from a change in internal state or the activity of a cell type in the network), or evidence accumulation (which again maps onto a cell type, or the activity of a network) - this model explains behavior via the stochastic sampling of a function of an exponential decay. The underlying model and the dynamics being simulated, as well as the process of stochastic sampling are well described, and the model fits the exponential function (equation 1) to data on a large array of worms exhibiting diverse behaviors (1600+ worms from Lopez-Cruz et al). The work of this study can explain or describe the inter-individual diversity of worm behavior across a large population. The model is also able to capture two aspects of the reorientations, including the dynamics (to switch or not to switch) and the kinetics (slow vs fast reorientations). The authors also work to compare their model to a few others including the Levy walk (whose construction arises from a Markov process) to a simple exponential distribution, all of which have been used to study foraging and search behaviors.

Weaknesses:

The weaknesses are one of framework, which may nonetheless stir discussion and motivate new ideas based on these results.

First, the examples the authors cite where a Gillespie algorithm is used to sample from a distribution, be it the kinetics associated with chemical dynamics, or a Lotka-Volterra Competition Model, there are underlying processes that govern the evolution of the dynamics, and thus the sampling from distributions. In one of their references for instance, the stochasticity arises from the birth and death rates, thereby influencing the genetic drift in the model. In these examples, the process governing the dynamics (and thus generating the distributions from which one samples) are distinct from the behavior being studied. In this manuscript, the distribution being sampled from is the exponential decay function of the reorientation rate. That the model performs well, and matches the data is commendable, but it is unclear how that could not be the case if the underlying function generating the distribution was fit to the data.

The second weakness is related to the first, in that absent an underlying mechanism or framework, one is left wondering what insight the model provides. Stochastic sampling a function generated by fitting the data to produce stochastic behavior is where one ends up in this framework. But if that is the case, what do we learn about how the foraging is happening. The authors suggest that the decay parameter M can be considered a memory timescale, which offers some suggestion, but then go on to say that the "physical basis of M can come from multiple sources". Here is where one is left for want: Molecular dynamics models that generate distributions can point to certain properties of the model, such as the binding kinetics (on and off rates, etc.) as explanations for the mechanisms generating the distributions, and therefore point to how a change in the biology affects the stochasticity of the process. It is unclear how this model provides such a connection.

The authors provide possible roadmaps, but where they lead and how to relate that back to testable mechanistic studies remains unclear. Weighing the significance of the finding relative to the weaknesses appears to depend on how one feels about the possible mechanisms the authors identify in their responses.

---

## [Author Response]

The following is the authors’ response to the previous reviews.

**Reviewer #2 (Public reviews):**
Weaknesses:This manuscript has two weaknesses that dampen the enthusiasm for the results. First, in all of the examples the authors cite where a Gillespie algorithm is used to sample from a distribution, be it the kinetics associated with chemical dynamics, or a Lotka-Volterra Competition Model, there are underlying processes that govern the evolution of the dynamics, and thus the sampling from distributions. In one of their references for instance, the stochasticity arises from the birth and death rates, thereby influencing the genetic drift in the model. In these examples, the process governing the dynamics (and thus generating the distributions from which one samples) are distinct from the behavior being studied. In this manuscript, the distribution being sampled from is the exponential decay function of the reorientation rate (lines 100-102). This appears to be tautological - a decay function fitted to the reorientation data is then sampled to generate the distributions of the reorientation data. That the model performs well, and matches the data is commendable, but it is unclear how that could not be the case if the underlying function generating the distribution was fit to the data.

To use the Lotka-Volterra model as an analogy, the changing reorientation rate (like a changing rate of prey growth) is tied to the decay in M (like a loss of predators). You could infer the loss of predators by measuring the changing rate of prey growth. In our case, we infer the loss of M by observing the changing reorientation rate. In the LotkaVolterra model, the prey growth rate is negatively associated with predator numbers, but in our model, the reorientation rate is positively associated with M, hence a loss in M leads to a decay in the reorientation rate.

You are correct that the decay parameters fit to the average should produce a distribution of *in silico* data that reproduce this average result (Figure 3a). However, this does not necessarily mean that these kinetic parameters should produce the same distributions of switch kinetics observed in Figure 3b. Indeed, a binary model (𝑴 ∈ {𝟎, 𝟏}), which produces an average distribution that matches the average experimental data (Figure 3a) produces a fundamentally different (bimodal) distribution of switch distributions in Figure 3b.

The second weakness is somewhat related to the first, in that absent an underlying mechanism or framework, one is left wondering what insight the model provides. Stochastic sampling a function generated by fitting the data to produce stochastic behavior is where one ends up in this framework, and the authors indeed point this out: "simple stochastic models should be sufficient to explain observably stochastic behaviors." (Line 233-234). But if that is the case, what do we learn about how the foraging is happening. The authors suggest that the decay parameter M can be considered a memory timescale; which offers some suggestion, but then go on to say that the "physical basis of M can come from multiple sources". Here is where one is left for want: The mechanisms suggested, including loss of sensory stimuli, alternations in motor integration, ionotropic glutamate signaling, dopamine, and neuropeptides are all suggested: this is basically all of the possible biological sources that can govern behavior, and one is left not knowing what insight the model provides. The array of biological processes listed are so variable in dynamics and meaning, that their explanation of what govern M is at best unsatisfying. Molecular dynamics models that generate distributions can point to certain properties of the model, such as the binding kinetics (on and off rates, etc.) as explanations for the mechanisms generating the distributions, and therefore point to how a change in the biology affects the stochasticity of the process. It is unclear how this model provides such a connection, especially taken in aggregate with the previous weakness.Providing a roadmap of how to think about the processes generating M, the meaning of those processes in search, and potential frameworks that are more constrained and with more precise biological underpinning (beyond the array of possibilities described) would go a long way to assuaging the weaknesses.

The insight we (hopefully) are trying to convey is that individual observations of apparent state-switching behavior does not necessarily imply that a state change is actually happening if a large fraction of the population is not producing this behavior. This same observation can be recreated by invoking a stochastic process, which we already know is how reorientation occurrences behave in the first place. Apparent switches to global foraging are simply due to the reorientation rate decaying in time, not necessarily due to a sudden state change. We modeled a stochastic binary switch (when M0=1) which produced a bimodal distribution of switch kinetics (Figure 3b) which was different than the experimental distribution.The biological basis of M is not addressed here, but we clarified the language on lines 342 and 343 to reinforce that it likely represents the timescales of AIA and ADE activities. We reiterated what was described in López-Cruz et al to convey that molecularly, what is governing the timescales of these two neurons is not trivial, and likely multi-faceted.

**Recommendations for the authors:**

**Reviewer #1 (Recommendations for the authors):**
The presentation of the Gillespie algorithm, though much improved, is tough going and for many biologists will be a barrier to appreciation of what was done and what was achieved. I found the description of the algorithm generated by AI (ChatGTP) to be more accessible and the example given to be better related to the present application of the algorithm. This might provide a template for a more accessible description of the model.

We are glad the newer draft is clearer, and apologize it is still difficult to read. We made a few changes that hopefully clarify some points (see below).

It is unclear how instances of >1 transition were automatically distinguished from instances with 1 transition. A related point is how the transition-finding algorithm was kept from detecting too many transitions, as it seems that any quadruplet of points defines a slope change.

In López-Cruz et al, >1 transitions (and all transitions) were distinguished by eye after running the findchangepts function. We added a clarifying statement on lines 78 and 79 to illuminate this point. As noted on line 72, the function itself only fits two regressions, so by definition, it can only define one transition. This is why we decided to plot the distribution of slope and transition parameters in the first place; to see if there was a clear bimodal distribution (as observed for other observably binary states, like roaming and dwelling). This was not the case for the experimental data, but was observed in the *in silico* data if we forced the algorithm to be a two-state model (Figure 3b, M0 = 1).

Line 113-4: I was confused by the distinction between the probability of observing an event and the propensity for it to occur. Are the authors implying that some events occur but are not observed?

We apologize for this confusion, and added some phrasing in Lines 115-130 to address this. The propensity is analogous to the rate of a reaction. Given this rate, the probability of seeing Ω+1 reorientations in the infinitesimal time interval dt is the product of the propensity and the probability the current state is Ω reorientations.

Line 120: Shouldn't propensity at t = 0 be alpha + beta?

Yes, thank you for catching this. We fixed it.

Why was it necessary to posit two decay processes (equations 2 and 5?). Wouldn't one suffice?

Thank you, we have added some text to clarify this point (lines 129-132). The Gillespie algorithm models discrete temporal events, which are explicitly dependent on the current state of the system. Since the propensity itself is changing in time, it implies that it is coupled to another state variable that is changing in time, i.e. another propensity. Since an exponential decay is sufficient to model the decay in reorientations, this implies that the reorientation propensity is coupled to a first order decay propensity (equations 4-5).

Line 145: ...sudden changes in [reorientation rate] are not due to...

Thank you, we have corrected this (Line 157).

Fig. 2d: Legend implies (but fails to state) that each dot is a worm, raising the question of how single worms with multiple transitions were plotted in this graph as they would have more than one transition point.

Thank you, we updated the legend. Multiple transitions are not quantified with the tworegression approach. Prior observations, such as by López-Cruz, were simply done by eye.

Line 153: Does i denote either process 1 or 2?

Yes, *i* is the subscript for each propensity *a_i_*. We have added text on line 166 to clarify this.

Line 159: Confusing. If an "event" is a reorientation event and a "transition" is a discrete change in slope of Omega vs t, then "The probability that no events will occur for ALL transitions in this time interval" makes no sense.

Thank you, we have reworded this part (Lines 169-172) to be clearer.

Equation 17:Unclear what index i refers to

Thank you, we have changed this to index to *j*, and modified the text on line 228 to reflect this.

Line 227-9: Unclear how collisions are thought to have caused the shift in experimental distribution.

We have clarified the text on lines 246 and 250. Collisions are not being referred to here, but instead the crossing of pheromone trails. This is purely speculative.

Line 310-317. If M rises on food, then worms should reorient more on food than after long times off food, when M has decayed. But worms don't reorient much on food; they behave as though M is low. This seems like a contradiction, unless one supposes instead that M is low on food and after long times off food but spikes when food is removed.

Thank you, we have added clarifying language on lines 333-336 to address this point. Worm behavior is fundamentally different on food, as worms transition to a dwell/roam behavioral dynamic which is fundamentally different than foraging behavior while off food.